# On the Quantization of AB Phase in Nonlinear Systems

**DOI:** 10.3390/e24121835

**Published:** 2022-12-16

**Authors:** Xi Liu, Qing-Hai Wang, Jiangbin Gong

**Affiliations:** 1NUS Graduate School—Integrative Sciences and Engineering Programme (ISEP), National University of Singapore, Singapore 119077, Singapore; 2Department of Physics, National University of Singapore, Singapore 117551, Singapore; 3Center for Quantum Technologies, National University of Singapore, Singapore 117543, Singapore

**Keywords:** AB phase, Berry phase, power-law nonlinearity, Dirac cone, adiabatic dynamics, quantization

## Abstract

Self-intersecting energy band structures in momentum space can be induced by nonlinearity at the mean-field level, with the so-called nonlinear Dirac cones as one intriguing consequence. Using the Qi-Wu-Zhang model plus power law nonlinearity, we systematically study in this paper the Aharonov–Bohm (AB) phase associated with an adiabatic process in the momentum space, with two adiabatic paths circling around one nonlinear Dirac cone. Interestingly, for and only for Kerr nonlinearity, the AB phase experiences a jump of π at the critical nonlinearity at which the Dirac cone appears and disappears (thus yielding π-quantization of the AB phase so long as the nonlinear Dirac cone exists), whereas for all other powers of nonlinearity, the AB phase always changes continuously with the nonlinear strength. Our results may be useful for experimental measurement of power-law nonlinearity and shall motivate further fundamental interest in aspects of geometric phase and adiabatic following in nonlinear systems.

## 1. Introduction

The dynamics depicted by a nonlinear discretized Schödinger equation (NDSE) can be extremely rich, including the emergence of many-dimensional chaos, solitons, and breathers, etc. The problem can be much reduced by assuming the translational invariance of a wave under consideration. With this assumption, the main physics is about the features of Bloch waves, the associated energy bands, and how they respond to changes in the parameters of a nonlinear system. Interestingly, the nonlinear Bloch bands of NDSE can induce gapless band structures absent in linear systems, such as 2-dimensional (2D) nonlinear Dirac cones [1] induced by Kerr nonlinearity [2]. Even more peculiarly, such nonlinear Dirac cones are formed by exotic nonlinear energy bands in a subregime of the Brillouin zone [1,3,4,5,6,7,8].

As a close analog to a setting in real space to measure the Aharonov–Bohm (AB) phase around a singularity point with magnetic flux, let us now imagine two adiabatic paths, in the momentum space, circling around a band-crossing point. If we adiabatically change the Bloch momentum, so as to guide the Bloch wave to evolve along the two adiabatic paths, the final phase difference thus generated between the two adiabatic paths is termed the nonlinear AB phase [1]. One may naïvely think of the following: provided that the dynamical phases between the two adiabatic paths are identical and hence have zero contribution to the phase difference of interest, the obtained AB phase would be just the Berry phase associated with the band degeneracy point. The actual physics turns out to be more interesting than just a Berry phase. As a result of nonlinearity, any small deviation of the adiabatically following state from the instantaneous Bloch wave causes a tiny correction to the dynamical phase, and accumulation of such tiny corrections over the entire adiabatic protocol yields an unfamiliar geometrical phase on top of the expected Berry phase. Remarkably, as a possible means of topological characterization of nonlinear Dirac cones, it is found in Ref. [1] that the nonlinear AB phase around nonlinear Dirac cones induced by Kerr nonlinearity added to the so-called Qi-Wu-Zhang (QWZ) model [9] is quantized in π, whereas the Berry phase is not quantized (thus in sharp contrast to a variety of linear systems, where the Berry phase around a Dirac cone is quantized in π [10,11,12,13]). Echoing with the finding in [1], Ref. [3] found π-quantization of a nonlinear Zak phase and Ref. [14] further confirmed the π-quantization of the nonlinear AB phase around a nodal line induced by Kerr nonlinearity.

The goal of this work is entirely focused on aspects of the nonlinear AB phase around Dirac cones induced by general power law nonlinearity [15,16,17,18,19,20,21,22,23]. In this way, it becomes possible to answer whether the previously obtained AB phase quantization is unique to Kerr nonlinearity and if so, why there is such uniqueness. Using the QWZ model [9] as the linear limit, we are able to analytically show that Kerr nonlinearity happens to be a critical case among all kinds of power law nonlinearity. Specifically, for any nonlinearity other than the cubic order, the π-quantization of nonlinear AB phase does not exist. Our analytical results are further confirmed by direct numerical simulations.

## 2. Hamiltonian and Energy Spectrum

The momentum-space Hamiltonian is composed of a QWZ model with power law nonlinearity characterized by a parameter *p*:(1)$$\begin{array}{cc} \hat{H}\left(\psi \right) & =J_{1}\sin k_{1}\sigma_{1}+J_{2}\sin k_{2}\sigma_{2}+\beta \left(k_{1},k_{2}\right)\sigma_{3}+g\left[\begin{array}{cc} |\psi_{1}{|}^{2p} & 0\\ 0 & |\psi_{2}{|}^{2p} \end{array}\right], \end{array}$$
where $\sigma_{i}$ are Pauli matrices and $\psi_{a}$ are two components of the wavefunction, $\psi =\left[\begin{array}{c} \psi_{1}\\ \psi_{2} \end{array}\right]$. The normalization of the wavefunction means that $|\psi_{1}{|}^{2}+{\left|\psi_{2}\right|}^{2}=1$. The nonlinearity parameter *p* is a non-negative real number. The Kerr nonlinearity corresponds to p=1. The parameters $k_{1}$ and $k_{2}$ are two quasimomenta, whose values will be adiabatically tuned in order to implement an actual adiabatic protocol to generate the nonlinear AB phase.

To solve the nonlinear eigenvalue problem,
(2)$$\begin{array}{cc} \hat{H}\left(\psi \right)|\psi \rangle & =E|\psi \rangle , \end{array}$$
we introduce a real parameter *x* as
(3)$$\begin{array}{cc} \psi_{1} & =\sqrt{\frac{1+x}{2}},\;\psi_{2}=\sqrt{\frac{1-x}{2}}e^{i\varphi }. \end{array}$$
We will see later that the angular variable φ is the same as in the Figure 1. It turns out that *x* is the central quantity for expressing energy, dynamical phase, Berry phase, and nonlinear AB phase. It can be shown that the instantaneous eigenenergy is
(4)$$\begin{array}{cc} E & =\frac{\beta }{x}+\frac{g}{x}\left[{\left(\frac{1+x}{2}\right)}^{p+1}-{\left(\frac{1-x}{2}\right)}^{p+1}\right], \end{array}$$
where *x* satisfies the following algebraic equation,
(5)$$\begin{array}{cc} \frac{1-x^{2}}{x^{2}}{\left\{\beta +\frac{g}{2}\left[{\left(\frac{1+x}{2}\right)}^{p}-{\left(\frac{1-x}{2}\right)}^{p}\right]\right\}}^{2} & ={\left|\gamma \right|}^{2}, \end{array}$$
with $\gamma :=J_{1}\sin k_{1}-iJ_{2}\sin k_{2}$.

In order to have a Dirac point in the energy spectrum, the energy must be doubly degenerate at $k_{1}=k_{2}=0$. Since γ=0 at this point, *x* must satisfy
(6)$$\begin{array}{cc} \beta \left(0,0\right)+\frac{g}{2}\left[{\left(\frac{1+x}{2}\right)}^{p}-{\left(\frac{1-x}{2}\right)}^{p}\right] & =0. \end{array}$$
For simplicity, we choose
(7)$$\begin{array}{cc} J_{1} & =J_{2}:=B, \end{array}$$
(8)$$\begin{array}{cc} \beta (k_{1},k_{2}) & =B(-1+\cos k_{1}+\cos k_{2}). \end{array}$$
Hence, β(0,0)=B. It is clear that the nonlinearity strength *g* and energy *E* can be scaled in terms of *B*. Energy spectra with p=1,1.5,2 and g=2.5B are shown in Figure 2, where the Dirac cone is clearly visible around the origin. A perturbative analysis of energy spectrum near the Dirac cone can be found in Section A.1.

## 3. Dynamics of Adiabatic Following

To obtain the nonlinear AB phase, let us consider two adiabatic paths along a small circle around the origin $k_{1}=k_{2}=0$. As shown in Figure 1, starting at the same point S, along each path, the system is guided to move along one half of the perimeter of the circle using the same amount of time. The two adiabatic paths are “recombined” at the end of the evolution at point N. As introduced in Section 1, the phase difference acquired by the system between two adiabatic paths is called the nonlinear AB phase. Clearly, the nonlinear AB phase here is the sum of the dynamical phase difference and the Berry phase associated with the closed loop around the band-degeneracy point. We shall study below the possible AB phase quantization for a varying nonlinearity strength *g* and for different nonlinear parameters *p*. The quasimomenta $k_{1}$ and $k_{2}$ associated with two spatial dimensions are parameterized by φ and will be made to adiabatically change.

At the starting point S, the system is assumed to be prepared in the Bloch eigenstate at momentum space location S. As the system adiabatically evolves along the path SEN or SWN, the time-evolving state deviates from the instantaneous eigenstate along the path, with the tiny deviation at the order of the adiabatic parameter ε. The slower the rate of adiabatic change is, the smaller ε is, and the smaller the deviation. Here, nonlinearity plays a key role. That is, the dynamical phase also obtains a correction at the order of ε. Since the total evolution time is of order $O(\varepsilon^{-1})$, the O(ε) term in this phase correction will contribute an ε-independent term through accumulation, yielding a geometric phase term out of the dynamical phase. This will not occur in linear terms because such correction accumulated over the entire adiabatic process is at most of the order of ε, which vanishes for sufficiently slow adiabatic protocols.

The dynamics of the states is governed by the time-dependent Schrödinger equation,
(9)$$\begin{array}{cc} i|\dot{\Psi }\rangle & =\hat{H}\left(\Psi \right)|\Psi \rangle , \end{array}$$
where the Hamiltonian is given by Equation (Equation 1) with ψ being replaced by Ψ. Here, the overhead dot denotes the time derivative. We will solve this equation up to the order of ε as described above. Through the lengthy computation, as illustrated in Section A.2, we obtain the instantaneous change rate of the overall phase of a time-evolving state as
(10)$$\begin{array}{cc} \dot{\theta } & ∼-E-\frac{1-x}{2}\dot{\varphi }+gp\frac{x(1-x^{2})}{4\Delta }\left[{\left(\frac{1+x}{2}\right)}^{p}-{\left(\frac{1-x}{2}\right)}^{p}\right]\dot{\varphi }, \end{array}$$
with
(11)$$\begin{array}{cc} \Delta & :=\beta +\frac{g}{2}\left[\left(1-px+px^{2}\right){\left(\frac{1+x}{2}\right)}^{p}-\left(1+px+px^{2}\right){\left(\frac{1-x}{2}\right)}^{p}\right]. \end{array}$$
We recognize that the circular integration of the second term in Equation (Equation 10) is nothing but the Berry phase $\theta_{B}$, because it assumes the same form as in the linear limit. The rest of the phase is from the dynamical phase $\theta_{D}$, which contains two parts: the first part comes from the instantaneous eigenenergy *E* and the second part from the third term in Equation (Equation 10) as a new contribution from the nonlinearity. Specifically,
(12)$$\begin{array}{cc} \theta_{B} & :=-\oint \frac{1-x}{2}d\varphi , \end{array}$$
(13)$$\begin{array}{cc} \theta_{D} & :=-\int Edt+gp\int \frac{x(1-x^{2})}{4\Delta }\left[{\left(\frac{1+x}{2}\right)}^{p}-{\left(\frac{1-x}{2}\right)}^{p}\right]d\varphi . \end{array}$$
In the event that the Dirac cone does exist at the point $k_{1}=k_{2}=0$, the obtained phase difference between the two adiabatic paths described in Figure 1 then becomes the nonlinear AB phase $\theta_{AB}$. Since the two adiabatic paths are symmetric by construction and they take the same amount of time, the leading term in Equation (13) contributes the same in each of the two paths. Thus, the difference of the dynamical phases between two paths comes from the second term of Equation (13) only. Thus, the total nonlinear AB phase is
(14)$$\begin{array}{cc} \theta_{AB} & :=\theta_{B}+\delta \theta_{D}\\ \; & ∼-\pi \left(1-x\right)+\pi gp\frac{x(1-x^{2})}{2\Delta }\left[{\left(\frac{1+x}{2}\right)}^{p}-{\left(\frac{1-x}{2}\right)}^{p}\right]. \end{array}$$

Note that we take into account that the paths are chosen to be close to the Dirac cone (so that the cones indeed have linear dispersion relations), namely, $|k_{1}|$ and $|k_{2}|$ are small at all times. The leading behavior of the dynamical phase difference term is then found to be
(15)$$\begin{array}{cc} \delta \theta_{D} & ∼\pi gp\frac{x_{0}\left(1-x_{0}^{2}\right)}{2\Delta_{0}}\left[{\left(\frac{1+x_{0}}{2}\right)}^{p}-{\left(\frac{1-x_{0}}{2}\right)}^{p}\right], \end{array}$$
where $\Delta_{0}$ is Δ evaluated at $x=x_{0}$ and $k_{1}=k_{2}=0$, $x_{0}$ is the solution of Equation (Equation 6), and
(16)$$\begin{array}{cc} \Delta_{0} & =-\frac{gpx_{0}}{2}\left[\left(1-x_{0}\right){\left(\frac{1+x_{0}}{2}\right)}^{p}+\left(1+x_{0}\right){\left(\frac{1-x_{0}}{2}\right)}^{p}\right]. \end{array}$$For the Berry phase, the leading behavior is
(17)$$\begin{array}{cc} \theta_{B} & ∼-\pi \left(1-x\right)∼-\pi \left(1-x_{0}\right). \end{array}$$

As detailed in Section A.1, For |g|>2B, a nonlinear Dirac cone is located at the origin. For |g|<2B, the only possible solutions to Equation (Equation 5) are x=±1 and there is no Dirac cone. For g∈(0,2B), we can hence assign $x_{0}=-1$, and for g∈(−2B,0), we may assign $x_{0}=1$. With this convention, it is clear to see that $\theta_{B}$ is constantly 0 (mod 2π) for g∈(−2B,2B). The Berry phase $\theta_{B}$ becomes nonzero and changes continuously for |g|>2B. For each *p*, as we continuously tune *g*, $x_{0}$ can be easily solved numerically using Equation (Equation 6), thus obtaining the theoretical values of the leading terms of the dynamical phase, Berry phase and AB phase around the origin. We also numerically solve the evolution using the Schrödinger equation Equation (Equation 9) along the two paths, and compute the dynamical phase, AB phase and Berry phase using numerical solutions of the evolution. The evolution is computed using an operator-splitting algorithm. The results are presented in Figure 3.

In each plot, solid lines are theoretical values, while dots on the solid lines are computed from numerical evolutions. In Figure 3a, for any p=0.5,1,1.5,2,2.5,3, the dynamical phase around the origin is 0 for g∈(−2B,2B). At the critical value g=±2B where the Dirac cone appears, for p=0.5, the Dirac cone changes continuously with respect to *g*. For p=1, there is a quantized jump of ±π at g=±2B. For p=1.5,2,2.5,3, there is a quantized jump of ±2π at the critical value g=±2B (so this is equivalent to no change). In Figure 3b, the Berry phase (modulo 2π) is identically 0 for g∈(−2B,2B), and changes continuously with respect to *g*. In Figure 3c, the AB phase (modulo 2π) is the sum of the dynamical phase in Figure 3a and the Berry phase in Figure 3b. Only for p=1, the AB phase has a quantized jump of π at the critical value g=±2B and stays at π for |g|>2B, as discovered by Ref. [1]. For all other values of *p*, the AB phase changes continuously with respect to *g*. The special behavior of p=1 is a result of the fact that p=1 is a critical value for the limit $\lim_{g\to \pm 2B^{\pm }}\delta \theta_{D}$, as will be explained in the next section.

## 4. Mechanism of the Jump of AB Phase at g=±2B for Kerr Nonlinearity

For p>1, we can factor out a factor $\left(1-x_{0}^{2}\right)$ from $\Delta_{0}$ which cancels the same factor in the numerator of $\delta \theta_{D}$,
(18)$$\begin{array}{cc} \delta \theta_{D}\left(p>1\right) & ∼-\pi \frac{{\left(1+x_{0}\right)}^{p}-{\left(1-x_{0}\right)}^{p}}{{\left(1+x_{0}\right)}^{p-1}+{\left(1-x_{0}\right)}^{p-1}}, \end{array}$$
which equals ∓2π or equivalently zero since $x_{0}=\pm 1$, for |g|=2B or when the Dirac cone starts to appear.

Likewise, for p=1, we have
(19)$$\begin{array}{cc} \delta \theta_{D}\left(p=1\right) & ∼-x_{0}\pi , \end{array}$$
which equals ∓π since $x_{0}=\pm 1$, for |g|=2B.

Finally, for 0<p<1,
(20)$$\begin{array}{cc} \delta \theta_{D}\left(0<p<1\right) & ∼-\pi {\left(1-x_{0}^{2}\right)}^{1-p}\frac{{\left(1+x_{0}\right)}^{p}-{\left(1-x_{0}\right)}^{p}}{{\left(1+x_{0}\right)}^{1-p}+{\left(1-x_{0}\right)}^{1-p}}, \end{array}$$
which vanishes for $x_{0}=\pm 1$, for |g|=2B.

The calculations above make it clear that the nonlinear AB phase associated with Kerr nonlinearity (p=1) is most special as the extra nonlinearity-induced correction to dynamical phase experiences a π jump when the Dirac cone appears. What is intriguing for Kerr nonlinearity is that the nonlinear AB phase stays quantized at π for |g|>2B, as $\theta_{B}$ and $\delta \theta_{D}$ happen to be complementary to each other, as shown in Equations (Equation 17) and (Equation 19). For all other forms of power-law nonlinearity, there is no such jump, π-quantization is thus absent, and consequently, the nonlinear AB phase only changes continuously with respect to *g*. This finally explains why in Figure 3 only the nonlinear AB phase for Kerr nonlinearity (p=1) displays a quantization plateau for |g|>2B.

## 5. Conclusions

In this paper, we analytically and computationally examined the so-called nonlinear AB phase around Dirac cones induced by power-law nonlinearity added to the QWZ model often used for studies of topological band structures. With our analytical results, we are able to explain why the nonlinear AB phase has a quantized jump of π when Dirac cone starts to appear or disappear, for and only for Kerr nonlinearity. In the context of nonlinear AB phase that can be in principle measured in experiments, Kerr nonlinearity is thus identified as a critical form of nonlinearity. As seen from our theoretical considerations above, our result will not be restricted to the QWZ model alone since it is based only on the asymptotic dispersion relation in the vicinity of the nonlinear Dirac cone. It is thus of considerable interest to investigate the generality of our results in other models with nonlinearity.

## Figures and Tables

**Figure 1 entropy-24-01835-f001:**
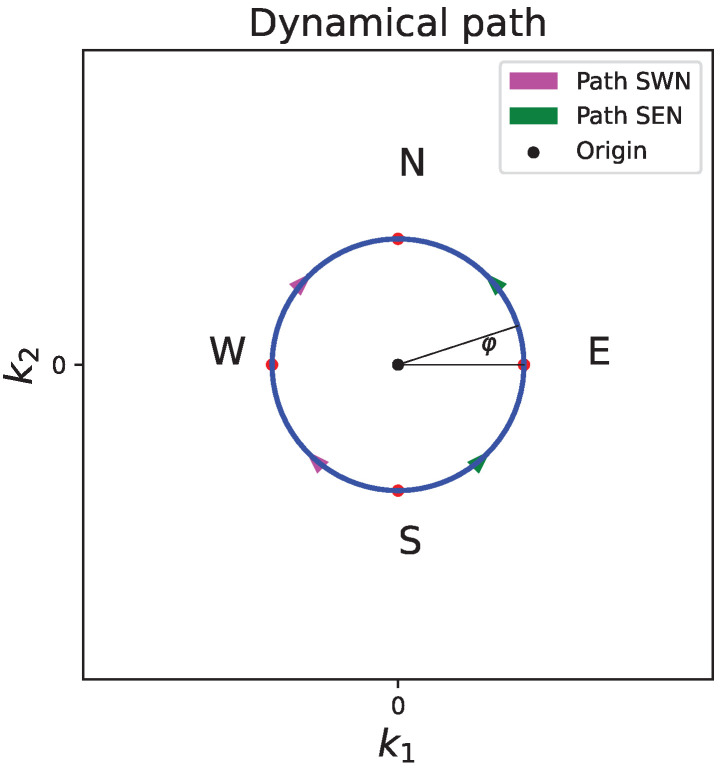
Dynamical paths in the momentum space. Path SWN and path SEN are symmetric halves of the perimeter of the circle. The system starts its adiabatic following at point S, and ends at point N. The two paths are parameterized by φ in the main text.

**Figure 2 entropy-24-01835-f002:**
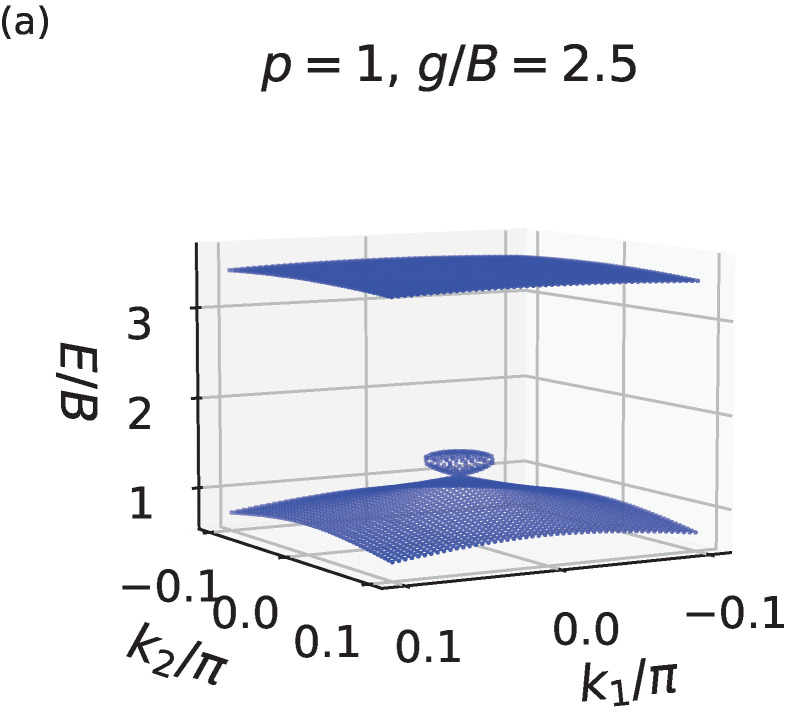
Nonlinear band structure for small momenta in the vicinity of the origin, i.e., for small values of $|k_{1}|$ and $|k_{2}|$, with the nonlinear strength parameter g=2.5B, and with the power-law nonlinearity parameter (**a**) p=1, (**b**) p=1.5, (**c**) p=2. See the main text for details of the system parameters. The Dirac cone emerges from the lower energy band.

**Figure 3 entropy-24-01835-f003:**
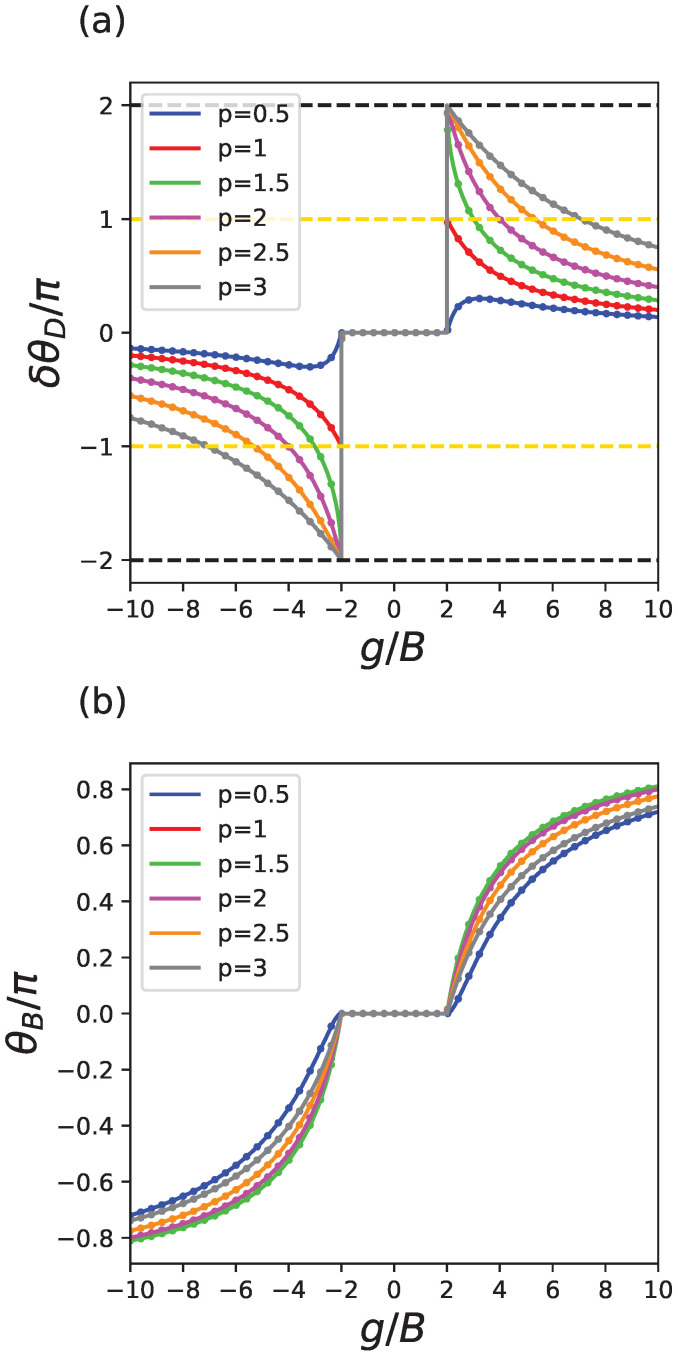
Dynamical, Berry and the nonlinear AB phase plotted against nonlinearity strength *g* for different values of power-law nonlinearity parameter *p*. The solid lines are theoretical values, and the dots are numerical verifications. (**a**) For p>1, the jump of dynamical phase at g=±2B is ±2π; for p=1, the jump of dynamical phase at g=±2B is ±π; and for p<1, the dynamical phase changes continuously. (**b**) The Berry phases change continuously for any *p*. (**c**) Only for Kerr nonlinearity p=1, the AB phase has a quantized jump of π at the critical value g=±2B and stays at π for |g|>2B.

## Data Availability

The codes for reproducing the results of this work can be found at https://github.com/saschapojot/diracCone (accessed on 16 November 2022).

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
