# Peer review of "On the Quantization of AB Phase in Nonlinear Systems"

_entropy, 2022, doi:10.3390/e24121835_

Round 1
Reviewer 1 Report
This is a very nice result. By adding a generic Kerr nonlinearity to the Qi-Wu-Zhang model used for investigations of topological band structure, the authors find that, only if the power of nonlinearity p=1, the AB phase is quantized. I would like to recommend the paper for publication, when the two issues below are taken into account properly:
1. I don’t understand how to formulate the “eigenvalue problem” for a nonlinear Schroedinger equation, and would like to invite the authors to educate me more, since this is the setup of their work. Looking at (4) what I can find is the relation between the “eigenenergy” E and the “eigenfunction” property, i.e. x, but NOT the spectrum. Is the latter important to their final statement, especially for g->\infty?
However, I do suspect this is NOT crucial to their work. As far as what I understood, what they would like to do is to seek for the solution of the form (3) to the “stationary” Schroedinger equation (2).
2. The conclusion is now based on QWZ model without nonlinearity. What happens if QWZ model is replaced by other models? Would a similar result follow?
